genomics; data; ethics; bioethics; screening

**Corresponding author:**
Anneke Lucassen,
E-mail: anneke.lucassen@well.ox.ac.uk

# Ethical issues raised by new genomic technologies: the case study of newborn genome screening

Rachel Horton[1,2,3] 🄳 and Anneke Lucassen[1,2,4] 🄳

[1]Centre for Personalised Medicine, St Anne's College, University of Oxford, Oxford, UK; [2]Clinical Ethics, Law and Society group, Wellcome Centre for Human Genetics, University of Oxford, Oxford, UK; [3]Wessex Clinical Genetics Service, Princess Anne Hospital, Southampton, UK and [4]NIHR Southampton Biomedical Research Centre, Southampton General Hospital, Southampton, UK

## Abstract

Over the last two decades, the ability to sequence a person's genetic code has improved exponentially, while the cost of doing so has plummeted. As genome sequencing is used more widely, diagnoses are being found for people with previously unexplained rare disease, and this has raised hopes that such analysis might usefully be employed to detect and mitigate diseases as early as possible in the life course. However, research with adults by initiatives such as population biobanks should shake our confidence in our ability to make clear health predictions from a genetic code – in many cases, we are learning that the links between genomic variants and disease are far less strong than we once thought. The UK Newborn Genomes Programme aspires to sequence up to 200,000 babies at birth, and analyse their genomic data aiming to identify '*actionable genetic conditions which may affect their health in early years. This aims to ensure timely diagnosis, access to treatment pathways, and enable better outcomes and quality of life for babies and their families*' (Genomics England, 2021). This is a laudable aim, but the path from obtaining genome sequences to enabling better outcomes will not be straightforward and illustrates many of the ethical challenges raised by the use of new genomic technologies. We focus particularly on the challenge of determining 'results' from the analysis of a genetic code, against a backdrop of promotional public discourses which tend to amplify best case scenarios from genome sequencing while minimising its potential to generate uncertainty.

## Impact statement

The UK Newborn Genomes Programme aims to sequence up to 200,000 babies at birth, analysing their genomic data in order to identify '*actionable genetic conditions that may affect their health in early years*'. The hope is that this will improve outcomes for these babies as prompt identification of disease will allow early and potentially more successful treatment. Our article discusses the difficulty of making clear health predictions from genetic code and the challenges that this may present for the Newborn Genomes Programme. Public discourses around newborn genomic screening often represent it as having the potential to give both breadth and clarity, but in reality, these are often in conflict – the more extensive the analysis, the greater the chance of generating uncertainty. Our article highlights the need to balance these discourses in order to allow meaningful consent conversations with parents considering enrolling their newborns in the Newborn Genomes Programme. While the programme will likely benefit society by deepening our understanding of the relationships between genetic variation and disease, the risk–benefit balance for the newborns themselves is less clear. Most babies will receive no benefit from taking part; a small number will substantially benefit from receiving a solid, treatable diagnosis that would not be detected with current screening; and depending on how the Newborn Genomes Programme chooses to balance sensitivity and specificity, a few will be embarked on lengthy and potentially specious diagnostic odysseys as we grapple with which genetic variants matter, and which we should ignore.

## Current UK newborn screening

The UK National Screening Committee currently recommends that all babies in the UK are offered newborn blood spot screening for nine rare conditions via a combination of biochemical and genetic screening on a dried blood sample ideally taken on day 5 of life. Parents should receive results by the time their baby is 6 weeks old (Public Health England, 2018). Studies interviewing parents about their decision-making around newborn screening indicate that screening is frequently portrayed and perceived as routine care for a newborn baby, rather than a choice over which parents might want to deliberate (Parsons et al., 2007; Nicholls and Southern,

2013). Most babies will receive a 'not suspected' result, and for the family, the experience of newborn screening may barely register. However, some babies will require a repeat sample before receiving a reassuring result, and a few will receive a 'suspected' result.

While a suspected result from newborn screening provides an opportunity for intervention and in some senses averts what otherwise might have been a lengthy diagnostic odyssey, such results often represent the start of a complex journey that will evolve over time (White et al., 2021). White et al. undertook a systematic review of 36 qualitative studies looking at parents' experiences of newborn screening and identified that screen-positive or inconclusive results '*ushered families into a compressed, critical window of time characterised by waiting periods, strong affective responses, and a need for more focused communication*', with effects rippling into the future. The authors write that their review '*underscores the need to move away from viewing newborn screening as a discrete series of clinical events and instead understand it as a process that can have far-reaching implications across time, space, and family groups*' (White et al., 2021).

### *The example of cystic fibrosis screening: Direct genetic screening comes at a cost*

Arguably the current screening test that leaves the most families with the greatest uncertainty is the one where direct genetic testing is most prominently involved in the testing pathway: cystic fibrosis. Currently, UK newborn blood spot screening for cystic fibrosis involves measuring immunoreactive trypsinogen (IRT), with analysis of *CFTR* (initially limited, then potentially more extensive) for babies with IRT levels over a specified cut-off. Babies with a 'CF suspected' result from newborn screening are referred to a regional cystic fibrosis centre for assessment, including a sweat chloride test, which if abnormally high will confirm the diagnosis. However, if sweat chloride is normal but the baby has two *CFTR* variants at least one of which has unclear phenotypic consequences, or sweat chloride is intermediate and the baby has one or no *CFTR* variants, they are said to have 'cystic fibrosis screen positive, inconclusive diagnosis' (CFSPID) (Public Health England, 2021). Many babies with CFSPID will not become unwell, but some will develop features of cystic fibrosis or a *CFTR*-related disorder at some point in the future (Munck, 2020).

Initially, the ongoing care that babies with CFSPID were offered was very variable, '*ranging from early discharge with little information to the family to full CF care in a CF centre*' (Munck et al., 2015), though in 2015 working groups supported by the European CF Society developed management recommendations based on a Delphi consensus approach (Munck et al., 2015). Boardman and Clark interviewed parents whose babies had CFSPID, finding that some parents managed the inherent uncertainty by thinking of their child as '*healthy*' or as having '*a rare form of CF*', while others described themselves as '*genetic nomads*' who did not fully belong in either the '*CF world, or the healthy kid world*' and oscillated between the two (Boardman and Clark, 2022). Screening for cystic fibrosis could be made more sensitive but the cost would be increasing the number of babies with CFSPID. A recent public dialogue on this topic found that while at the start most participants advocated an approach that would minimise the number of true cystic fibrosis cases missed, following discussion most participants came to prefer a more specific approach that minimised the number of CFSPID cases. Most participants '*struggled with the moral dilemma presented by the outcomes of the two approaches*' but '*highly prized certainty of outcome in screening linked to clear actions to improve health when compared with approaches that could result in less clarity or long-term uncertainty*' (Kinsella et al., 2022).

### Genome screening for newborns

The UK currently takes a conservative approach to newborn screening (e.g., in the US, the Recommended Uniform Screening Panel for newborns includes 20 metabolic conditions on its core panel (Advisory Committee on Heritable Disorders in Newborns and Children, 2020), in contrast to the UK's six). The Genetic Alliance report *Fixing the present, building for the future* makes a compelling case that some criteria used by the UK National Screening Committee to determine what conditions are screened for were not developed with rare disease in mind, and highlights that '*for [families] that might have received [an] early warning had their child been born in a different country, the urgency to fix newborn bloodspot screening is acute*' (Genetic Alliance, 2019). The Newborn Genomes Programme now plans that '*building on the principles of the NHS newborn screening programme, up to 200,000 babies' genomes will be sequenced and analysed for a set of actionable genetic conditions which may affect their health in early years*' (Genomics England, 2021). There are major benefits to making solid diagnoses early where good treatments are available, and where genetic screening can achieve this, we should use it. However, the question as to what extent genomic screening will be capable of doing this needs urgent attention, and public discourses around newborn genomic screening should reflect the limitations and potential pitfalls as much as the aspirations.

### *What qualities should we demand of newborn genome screening results?*

Genome sequencing catalogues the millions of variants present in a person's genetic code. Filters then sift out variants that might be relevant to whatever question is being asked. Choosing filters to interrogate the genomic data of a 'healthy' newborn is challenging – what questions should we ask regarding a baby's future health, and what qualities should we demand of the answers? When and why should we consider a particular genetic variant as constituting a 'result'?

Each person has over four million genomic variants, which could be sifted to generate information ranging from where their recent ancestors might have lived, to predispositions to adult-onset conditions. Most variants will have no known impact on health, but many could appear concerning based on hypothetical data (Ghosh et al., 2017). A 2016 analysis found that the average ExAC participant had around 54 variants reported as disease-causing in two widely-used databases of disease-causing variants – most were thought to be misclassified in these databases, reflecting the imperfect and evolving state of understanding around genomic variation (Lek et al., 2016). A recent analysis of UK Biobank participants showed that when using a panel to look at more than 500 disease genes, most people had one or more rare non-synonymous variants (Beaumont and Wright, 2022). Although these variants looked hypothetically concerning, the context in which they were found (mainly healthy adults over the age of 45) meant that they were likely to be benign. However, if such variants were to be found in newborn babies, who might not yet have had time to develop associated symptoms and signs of the relevant disease, they would be harder to dismiss.

Even for variants where an association with disease is strongly established, their medical consequences might be different when found outside the context of a personal or family history of the disease in question (Kingdom and Wright, 2022). For example, the *HNF4A* c.340C>T variant was thought to have a penetrance of up to 75% at 40 years, based on a study of large maturity-onset diabetes of the young (MODY) cohort. Wright et al. (2019) analysed the same variant in UK Biobank participants and found that under 10% of people with the variant had developed diabetes by age 40.

The Newborn Genomes Programme vision document predicts that as a conservative estimate, '*every year, 3000+ babies could benefit from life-saving or life-changing interventions thanks to [whole genome sequencing]*' (Genomics England, 2021). The Newborn Screening in Genomic Medicine and Public Health (NSIGHT) programme in the US has been exploring the role of genome sequencing in newborn screening since 2010 (Holm et al., 2018; Roman et al., 2020) and illustrates the complexities inherent in attempts to bridge from a newborn's genome sequence to treatment recommendations.

One NSIGHT project, NC NEXUS, enrolled 106 babies for exome sequencing. A 466 gene panel was used to examine their exome data (17 babies had a previously diagnosed metabolic condition, and 28 had hearing loss – these babies also had analysis relating to these indications) (Roman et al., 2020). Four out of the 106 children had a 'positive' screen on the 466 gene panel (not including findings that would explain their reason for recruitment-metabolic problems or hearing loss) (Roman et al., 2020). The study was said to have '*implications for [the children's] health supervision*' with the suggestion that '*expanding the current [newborn screening] panel to include other actionable conditions detectable only by sequencing could further enhance the public health benefits of [newborn screening]*', though the authors noted the challenges in balancing case detection against false positives (Roman et al., 2020). Table 1 summarises the findings, highlighting that 'actionable' findings often had uncertain significance or implications for the child.

The BabySeq project was another US-based study that offered 159 newborns exome sequencing in addition to conventional newborn screening (127 were considered healthy; 32 were enrolled from intensive care units). The project aimed to report variants that indicated risk, or carrier status, for highly penetrant childhood-onset conditions (Holm et al., 2018). Parents could also opt to hear about variants relating to actionable adult-onset disease risks, and did so for 85 of the babies. Fifteen newborns had a variant thought to be associated with childhood-onset disease (10 were apparently healthy newborns; 5 had been enrolled from intensive care and were already symptomatic) (Ceyhan-Birsoy et al., 2019). Table 2 summarises the variants found in the newborn-screened babies.

Reflecting on the NSIGHT experience, the NSIGHT Ethics and Policy Advisory board called for nuanced use of genome sequencing in newborns. They recognised that genome sequencing can be extremely valuable for unwell babies but suggest that while there is '*considerable benefit in using targeted sequencing to screen for … conditions that meet the criteria for inclusion in newborn screening panels, use of genome-wide sequencing as a sole screening tool for newborns is at best premature*' (Johnston et al., 2018).

### What expectations might people have from newborn genome screening?

Those developing analytic pipelines for the Newborn Genomes Programme will be well aware of the uncertainties associated with interpreting genetic code in 'healthy' babies, but public-facing discussion around this initiative tends to focus on its aspirations more than its probable limitations. For example, the Newborn Genomes Programme vision document envisages how '*providing [whole genome sequencing] for newborns could help transform diagnostic odysseys … ensuring babies get access to appropriate treatments and interventions much earlier. It could also enable researchers to discover and develop new ways to use genomic medicine to help treat and save lives – and it could usher in a future of personalised, preventative healthcare*' (Genomics England, 2021). In 2019 a former UK Health Secretary announced his ambition '*that eventually every child will be able to receive whole genome sequencing along with the heel prick test. We will give every child the best possible start in life by ensuring they get the best possible medical care as soon as they enter the world. Predictive, preventative, personalised healthcare – that is the future of the NHS – and whole genome sequencing and genomics is going to play a huge part in that*' (Blanchard and Hurfurt, 2019).

**Table 1.** Actionable findings from the NC NEXUS study

| Gene | Condition of concern | Comments |
| --- | --- | --- |
| *LDLR* | Familial hypercholesterolaemia | • Might improve outcomes, as children with familial hypercholesterolaemia may benefit from lipid monitoring and treatment during childhood<br>• Parents already aware of family history of hypercholesterolaemia and the child could instead have relevant testing at an age where it might change management |
| *OTC* | OTC deficiency (X-linked metabolic condition) | • Subsequent metabolic screening of child normal and variant found in unaffected grandfather<br>• Younger brother was tested for variant *in utero* and monitored for OTC deficiency from birth, but remained asymptomatic at the time of reporting. Possible that monitoring will avert severe illness, but perhaps more probable that the boy would stay healthy like his grandfather |
| *DSC2* | Arrhythmogenic right ventricular dysplasia (heart condition) | • Initial echocardiogram normal but cardiology follow-up recommended as the condition can develop over time<br>• Knowing that a baby has this variant might mean a heart condition that does develop is detected and treated early, but might also mean a lifetime for screening for a condition that will never happen |
| *F11* | Autosomal recessive factor XI deficiency (bleeding disorder) | • Child had nosebleeds and was referred to a haematologist<br>• If factor XI deficiency is confirmed (unknown at the time of reporting), this would be helpful for managing future injuries or planning medical procedures |

**Table 2.** Actionable childhood-onset disease risk determined in apparently healthy babies in the BabySeq project

| Gene | Condition of concern | Comments |
|---|---|---|
| *BTD* | Biotinidase deficiency (metabolic disorder) | • Subsequent biochemical testing indicated partial biotinidase deficiency<br>• People with partial biotinidase deficiency can develop mild hypotonia, skin rash and hair loss, particularly during times of stress. Unclear whether baby would develop symptoms if untreated, but treatment (lifelong oral biotin) is cheap with no known side effects (Murry *et al.*, 2018) |
| *CD46* | Atypical haemolytic uraemic syndrome | • Genetic variant inherited from mother<br>• Likely confers a predisposition to developing atypical haemolytic uraemic syndrome, rather than directly causing it<br>• The finding might allow avoidance of known triggers but the chance of the baby developing atypical haemolytic uraemic syndrome at some point in their life cannot be quantified |
| *ELN* | Supravalvular aortic stenosis | • Genetic variant inherited from father<br>• Cardiac review for baby (and father) could check for asymptomatic stenosis and allow treatment |
| *KCNQ4* | Non-syndromic hearing loss | • Inherited from a parent (grandparent with hearing loss)<br>• Baby had no issues on newborn hearing screen, but may go on to develop hearing loss. Chance of this cannot be quantified<br>• Knowing this variant allows parents to be vigilant for possible hearing loss as the child grows and mean they can be careful about, for example, reducing exposure to loud noise, but hearing loss may not happen for many years, if at all |
| *MYBPC3* | Hypertrophic cardiomyopathy | • Variant inherited from mother<br>• Not all those with the variant will develop cardiomyopathy, and may not develop until adulthood, but the mother and baby could have long-term cardiac follow-up aiming to detect cardiomyopathy early |
| *TTN* (in four babies) | Dilated cardiomyopathy | • All variants had been inherited from a parent<br>• Some evidence of penetrance and age of onset from a UK Biobank analysis which estimated that for 10,000 people aged around 64 years with a truncating variant in *TTN*, 340 would already be known to have cardiomyopathy, and one-off cardiac imaging would detect around 240 more people with cardiomyopathy. For the 9,420 with a *TTN* variant but with normal cardiac imaging, for every 8,000 person years of serial imaging (1,600 cardiac MRI scans) there would be 25 new cardiomyopathy diagnoses and the opportunity to prevent one death over the following 4 years (McGurk *et al.*, 2022)<br>• Lifelong cardiac follow-up for these babies and their parents might reassure that if dilated cardiomyopathy develops, it will be picked up early, but this is resource-intensive and in many, the condition will never manifest |
| *VCL* | Dilated cardiomyopathy | • Variant inherited from mother<br>• *VCL* variants may act as genetic modifiers that can precipitate or lead to more severe disease in the context of other risk factors (genetic and/or environmental) (Hawley *et al.*, 2020)<br>• Lifelong cardiac follow-up for this baby and her mother might detect dilated cardio-myopathy early, but the condition might never occur |

As evidenced in Tables 1 and 2, many of the results that will be found via newborn exome or genome screening are more nuanced than the above quotations suggest. In the NC NEXUS and BabySeq projects, findings often resulted in specific but difficult-to-quantify risks that resource-intensive surveillance might go some way to address, rather than clear-cut diagnoses with effective treatments. While having had exome screening via NC NEXUS or BabySeq may well turn out to have influenced the health of babies and families in positive ways, it will take decades until we know which families benefited, and which invested time, money and anxiety attempting early detection of diseases that never occurred. Promotional discourses around newborn genome screening risk generating expectations that such screening will bring both breadth and clarity where in reality these are often in conflict – the wider we look, the more uncertainty we invite. These discourses need balancing to create an environment where meaningful consent conversations can take place regarding participation in the Newborn Genomes Programme.

Clearly, accurate public representation of possible results for participants in newborn genome screening will be contingent on analytic decisions which are yet to be made. If the threshold for declaring results were to be similar to those in the NSIGHT project, we need to publicise how many more babies and families will be left with uncertainty from newborn screening, and demand answers as to whether the NHS has sufficient resources to actually deliver the subsequent surveillance that might be recommended based on the results. If more conservative analysis is planned, we need to publicise the rationale for collecting whole genomes when far more limited testing would be capable of providing the same information to babies and their families. We have to be careful not to imply that newborn genome screening is essentially the current heelprick test, but better.

We should also consider how the offer of newborn genome screening might impact on the uptake of current screening. Currently, the standard heelprick test in the UK is often presented as routine care, and although parental consent could be withheld, it is often not routinely sought in any depth since participation is implicitly expected in the baby's best interests. In some countries, for example, the USA, the best interests justification for testing is more explicit and newborn screening is mandatory with few

options to opt out (McCandless and Wright, 2020). It is feasible that some parents who might have agreed to the standard heelprick test when presented as part of routine care, may opt-out of screening altogether if they are instead essentially asked to decide what level of newborn screening they would like. That is to say that, if newborn genome screening is only possible via more formalised parental consent processes, it is possible that one cost of introducing such screening will be that more babies get no screening at all.[1]

### How might newborn genome screening impact on NHS resources?

The resource implications of newborn genome screening decisions will be significant. In discussing the potential for genomics to predict disease, the *Genome UK* report outlines that '*Effective disease prevention benefits not just the individual but the healthcare system as a whole. We know that waiting until a patient presents to hospital with a condition leads to worse health outcomes and increased care costs*' (Department for Health and Social Care, Department for Business et al., 2020). However, this applies to situations where conditions will actually one day manifest – for some of the babies (and parents) in the NSIGHT studies they may undergo a lifetime of costly screening but never develop the condition in question.

The proposed UK pilot will screen up to 850 times more babies than the 'healthy babies' screened by NC NEXUS and BabySeq combined, with a broader technology (genome rather than exome sequencing). If a similar threshold for 'actionable findings' were applied as to those for the NSIGHT projects (i.e., 3–8% of babies end up being referred for specialist review or investigations that might need repeating at intervals throughout their life), this would dramatically increase the burden on the publicly funded NHS. In the long term, some of these costs may be recouped if interventions successfully prevent people from developing fulminant disease, but the costs of long-term follow-up in babies who would never go on to develop the condition in question will likely be significant.

A key concern is how to ensure that offering genome screening to large numbers of 'well' babies does not detract from the care provided to the people living with rare disease that the Newborn Genomes Programme aspires to help. Genome sequencing has an excellent track record when deployed to answer specific clinical questions about unwell newborns: many studies show that exome and genome sequencing in neonatal and paediatric critical care settings is effective at finding diagnoses for unwell children, and can usefully influence medical care (Clark et al., 2018; Mestek-Boukhibar et al., 2018; Chung et al., 2020; Dimmock et al., 2020; Krantz et al., 2021). However, many people with rare conditions remain on a diagnostic odyssey, despite having had genome sequencing – for example, 75% of probands in the 100,000 Genomes Project do not yet have a molecular diagnosis for the health issues that led them to join the project (Smedley et al., 2021).

We need to optimise diagnostic pathways for people with existing health issues, and ensure that people with rare disease have access to timely diagnostic genomic tests. The drive to introduce newborn genome screening must be tempered by the capabilities of current systems to cope with the output. If screening 'healthy' babies silts up genetic laboratories and so delays analysis of genome tests for sick babies, or adds to the pressure on services that already have long waiting lists, it will do no favours for the people affected

by rare conditions that it hopes to benefit. This is not to say that newborn genome screening cannot be of benefit, but the investment around it must anticipate the far-reaching downstream consequences of identifying a suspected genetic diagnosis, and ensure that the services that will be called into action are adequately resourced, or the enterprise may hinder more than it helps.

### Why does the newborn programme propose whole genome screening?

Given the scale of the project, it seems likely that at least initially the Newborn Genomes Programme will take a conservative stance and aim to report only well-understood variants in very well-understood genes, trying to maintain a similar balance between diagnoses and uncertainty to that of the current screening programme. If this is the case then the genetic data necessary for such screening could be obtained much more efficiently than by sequencing a baby's entire genome. The rationale for genome sequencing therefore needs to be made very clear to parents.

As discussed earlier, it is unclear what many genetic variants might mean for a person's health over the course of their life, and ultimately it is by collecting these data on a large scale that we will learn more. The Newborn Genomes Programme stands to greatly contribute to our understanding of genetic variation and its consequences by developing a database of genomes linked over time to the health problems that babies go on to experience. Many parents may be happy to contribute their child's data to this on altruistic grounds, but it needs to be explicit that genomic data (as opposed to more limited genetic data) is primarily being collected to form a database that will have major scientific (and monetary) value, rather than because the genome of each specific child will be analysed in depth to give them '*the best possible start in life by ensuring they get the best possible medical care as soon as they enter the world*'. For the babies themselves, most will receive no benefit from participating; a small number will substantially benefit from receiving a solid, treatable diagnosis that would not be detected with current screening; and depending on how the project chooses to balance sensitivity and specificity, a few will be embarked on lengthy and potentially specious diagnostic odysseys as we grapple with which genetic variants matter, and which we should ignore.

### Consent for newborn genome screening

The Newborn Genomes Programme has identified '*person-centred consent across screening, research and re-analysis*' as a theme to prioritise in the co-design and feasibility phase, explaining that: '*We need to set the bar to ensure all parents are empowered to make informed choices in terms of opting into the programme and understanding what it entails – and how their children will be able to make their own choices too*' (Genomics England, 2021). There will be many challenges to navigate in ensuring that consent is as robust as possible.

Clearly, newborns will be incapable of engaging with the pros and cons of genomic screening and decisions will need to be made on their behalf. Consent will be asked of their parents but how to time conversations in order to maximise parents' ability to engage with the relevant issues is challenging. The immediate postpartum period is not conducive to making complex decisions with lifelong consequences, and evidence regarding participation in the current newborn screening programme shows that this often barely registers as a decision (Nicholls and Southern, 2013). During pregnancy, people may have more time to weigh up the pros and cons of genomic testing, though the future person for whom they are

---

[1]Thanks to an anonymous reviewer for raising this issue.

making decisions may be less tangible. However engaged people are, there are also major challenges in forecasting the likely consequences of participation for a newborn, both in terms of what the short-term results may look like, but also whether there might be later consequences (positive or negative) to having their genomic data catalogued. This does raise the question as to why newborns have been selected as the target population for genome screening as opposed to, for example, young adults, who could make these decisions for themselves. One argument would be that newborns stand to gain the most from the analysis as many rare diseases affect children and earlier diagnosis means earlier treatment, though as discussed previously the journey from sequencing a 'healthy' baby's genome to improving outcomes for them may be convoluted.

Another aspect of consent, that is common to other genetic tests, is that testing a newborn will also in part test their biological parents and siblings. For example, many of the variants identified in NSIGHT babies had been inherited from a 'healthy' parent, and often the genetic predisposition was at that time potentially more relevant to the parent's health than to their baby's, particularly where they had elected to learn about their child's adult-onset disease risks. The provision of indirect information on parents may be useful, for example, the BabySeq project noted that '*identification of a newborn's carrier status can facilitate parental testing and reproductive planning for the family*' (Holm et al., 2018). Helping parents to mitigate their own disease risks can also be considered to be in a baby's best interests when taking a holistic view. However, there are difficult questions to consider as to how far to stretch the power to make inferences about parental health from a baby's genome. Babies are a captive audience – they will be entirely dependent on the choices that their parents and society make for them, though they will have to live with the consequences. In families with known adult-onset genetic risks, people are generally not offered testing until they are old enough to make their own decision as to whether or not they want it. The further we stray into commenting on adult-onset risks or carrier status, the more we need to consider whether our target population should be different. Preconception carrier screening or adult genome screening could achieve similar effects and would not require the involvement of people who cannot say no.

The 100,000 Genomes Project has some overlap with the Newborn Genomes Programme in terms of the issues needing discussion with prospective participants, as this project involves genomic testing though in this case aiming to find diagnoses for rare disease, or to improve cancer care. Research exploring 100,000 Genomes Project participants' experiences found that some held misconceptions about the project and some could not recollect the decisions they had made as part of the consent process (e.g. whether or not to have additional findings). However, people reported being satisfied with their experience and seemed unconcerned that they could not remember some details of what they consented to – they trusted that health professionals and the project would act in their interests (Ballard et al., 2020). The Newborn Genomes Programme is likely to benefit from similar trust when recruitment starts, but it is important to consider the potential fragility of this trust: a programme that changes parents' perception of their baby from healthy to sick (albeit with the motivation of allowing early care and better outcomes) will likely be more vulnerable than a programme trying to find explanations for people who are already experiencing health problems.

While our article has focussed on the Newborn Genomes Programme, many of the points we discuss relate to newborn genome screening in general, rather than this programme in particular. We recognise that people developing the Newborn Genomes Programme are in a challenging situation – as their plans (appropriately) involve extensive consultation, they are not yet crystallised and so the rhetoric around them will naturally focus on their aims rather than their detail. The relative importance of the issues we raise cannot be clear until the details of the project are more developed. For example, if the project will involve extensive analysis of each baby's genome the generation of uncertainty will be highly relevant; if more conservative analysis is planned the collection of data well beyond what is needed to undertake the analysis may be more pertinent. The nebulous nature of current plans gives developers a real opportunity to respond to concerns raised as they develop the project. Here, we outline issues that need consideration, and argue that if consultations are to be meaningful, they must put these firmly under the spotlight.

## Conclusions

The ability to sequence genomes quickly and easily has many benefits, but their use to inform clinical care also brings challenges, many of which are well illustrated by the case of newborn genome screening. Newborn screening programmes must tread a delicate balance between identifying as many babies with serious but treatable conditions as possible, and not raising unjustified anxiety around healthy babies. Genome sequencing could screen for far more conditions than conventional screening, but in tandem with more diagnoses would come more false positives, and uncertain findings where it may take a lifetime of potentially costly screening to be sure that the condition of concern will never manifest. These aspects are often missing from public discourses around newborn genome screening, inviting people to expect that genome screening will bring both breadth and clarity when in reality choices have to be made as to which to prioritise. Sequencing healthy newborns may benefit society in the longer run, by creating a research resource that enables us to learn how our genetic code influences our health over time. However, whether, on average, the newborns themselves are likely to benefit is not yet clear.

**Open peer review.** To view the open peer review materials for this article, please visit http://doi.org/10.1017/pcm.2022.2.

**Acknowledgements.** Thank you to Dr Caroline Wright for her very constructive and thought-provoking comments on the first draft of this article.

**Financial support.** R.H.'s work is funded by a Wellcome Trust Research Award for Health Professionals in Humanities and Social Science 218092/A/19/Z. A.L.'s work is supported by funding from a Wellcome Trust collaborative award 208053/B/17/Z. This research was funded in whole, or in part, by the Wellcome Trust (218092/A/19/Z and 208053/B/17/Z). For the purpose of open access, the author has applied a CC BY public copyright licence to any Author Accepted Manuscript version arising from this submission.

**Competing interests.** The authors declare no competing interests exist.

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
