## [Reviewer Report]

*Comments to Author*: This paper raises important issues around the potential use of whole genome sequencing in screening and diagnosis of genetic conditions. The authors correctly highlight the importance of balancing the possible benefits with uncertainty within public discourse and consent conversations with parents when considering participation of their newborns in the Newborn Genomes Programme. In my review of the paper, I have shared ways in which the Newborns programme team is taking a considered approach to developing the consent process and conversation with parents to include considerations around uncertainty when they are approached about the study. The authors also argue that the majority of babies will derive no benefit from participation and that the primary aim of the programme is to develop a research resource over time. It is true that most babies will not directly receive positive findings from the initial panel of actionable conditions, since the programme is focused on rare, actionable, childhood-onset conditions, which affect a relatively small number of babies. We anticipate 500-1000 babies from the cohort of 100K will receive positive findings with direct benefit for their health. However, this does not take away from the primary and most important aim of the programme which is to find and support newborns with treatable conditions. The newborns programme team is currently deliberating over decisions around the consent model for the programme with external experts, and discussing whether the core part of the programme (screening research) ought to be separated from parents consenting for their babies genomes to be used for wider research, making the latter an optional additional for parents who wish to participate in the study. Overall, the article raises important factors to consider when designing and evaluating this research study, though it is not always clear whether the arguments presented pertain to doing the study itself vs. precautions if whole genome sequencing were to become part of a future service within newborn screening. The paper could be strengthened by being explicit about where the authors feel that the Newborn Genomes Programme is not adequately addressing these concerns as part of its development vs. where the issues raised are more broadly relevant to genetic screening programmes or newborn screening. Further comments are attached in the document.

---

## [Reviewer Report]

*Comments to Author*: This is an excellent, and very timely article. It describes the ethical issues raised by the proposed large UK study of whole genome sequencing in neonates.

While acknowledging that early diagnosis in symptomatic babies can be helpful, it highlights all the disadvantages of the discovery of uninterpretable variants or false positives.

The importance of very long term follow-up, in order to assess the impact of the proposed programme on babies and their parents, is explained very clearly.

The paper also identifies the risk to other NHS services of introducing such a large programme when labs are already struggling to deal with diagnostic and predictive tests in other situations, and specialist clinics are full of affected individuals, leaving no time for them to accommodate screen positive babies for whom additional investigations and counselling may be required, even though the babies may never develop any signs of the disease in question.

---

## [Reviewer Report]

*Comments to Author*: The article "Ethical issues raised by new genomic technologies: the case study of newborn genome screening" by Horton et al. certainly merits publication. Based on their research and knowledge of similar NBS-WGS efforts in the USA, as well as lessons learned from the introduction of WGS in the UK's 100,000 Genomes Project (adults), the authors rightly urge caution if not a "rethink" of the proposed Newborn Genomes Programme (NGP). As pointed out by the authors, neither the rationale of the NGP (i.e., "best possible start in life"), nor the strategy for handling the long-term unforeseen consequences of the NGP are justified or addressed.

My sole concern with the article, however, is that the authors do not address the possible impact on the routine, standard of paediatric care that NBS has become. As the most successful public health programme in history, NBS has become the right of the at-risk asymptomatic newborn to be found and treated. If individual, written (?) parental consent as opposed to current parental notification of NBS is to become the new "normal", there is no doubt that the universality of this programme will be affected as an explicit consent would have to be obtained as it is now for the use of WGS in neonatal medical care. From a health systems point of view, from a logistics point of view, from the rights of the child point of view, who will be losing, not gaining?

---

## [Reviewer Report]

*Comments to Author*: Thank you to the authors for their comments and sharing a revised version of the paper in light of reviewer feedback.

---

## [Reviewer Report]

*Comments to Author*: The authors have responded effectively to the comments of the reviewers, and have added in 2 particularly useful references, which have recently been published.